# Profiling the Urobiota in a Pediatric Population with Neurogenic Bladder Secondary to Spinal Dysraphism

**DOI:** 10.3390/ijms24098261

**Published:** 2023-05-05

**Authors:** Flavio De Maio, Giacomo Grotti, Francesco Mariani, Danilo Buonsenso, Giulia Santarelli, Delia Mercedes Bianco, Brunella Posteraro, Maurizio Sanguinetti, Claudia Rendeli

**Affiliations:** 1Dipartimento di Scienze di Laboratorio e Infettivologiche, Fondazione Policlinico Universitario A. Gemelli IRCCS, 00168 Rome, Italy; 2Dipartimento di Scienze della Salute della Donna, del Bambino e di Sanità Pubblica, Fondazione Policlinico Universitario A. Gemelli IRCCS, 00168 Rome, Italy; 3Centro di Salute Globale, Università Cattolica del Sacro Cuore, 00168 Rome, Italy; 4Dipartimento di Scienze Biotecnologiche di Base, Cliniche Intensivologiche e Perioperatorie—Sezione di Microbiologia, Università Cattolica del Sacro Cuore, 00168 Rome, Italy

**Keywords:** spina bifida, clean intermittent catheterization, urobiota

## Abstract

The human bladder has been long thought to be sterile until that, only in the last decade, advances in molecular biology have shown that the human urinary tract is populated with microorganisms. The relationship between the urobiota and the development of urinary tract disorders is now of great interest. Patients with spina bifida (SB) can be born with (or develop over time) neurological deficits due to damaged nerves that originate in the lower part of the spinal cord, including the neurogenic bladder. This condition represents a predisposing factor for urinary tract infections so that the most frequently used approach to treat patients with neurogenic bladder is based on clean intermittent catheterization (CIC). In this study, we analyzed the urobiota composition in a pediatric cohort of patients with SB compared to healthy controls, as well as the urobiota characteristics based on whether patients received CIC or not.

## 1. Introduction

The term microbiota refers to microbes that live inside or on the surface of the human body, while the term microbiome denotes the collection of genes and their products from these microbes. In 2008, the Human Microbiome Project (HMP) was launched with the purpose of characterizing the human microbiome and analyzing its role in human health and disease. Since the urinary microbiome (microbiota) was not part of the project, urine in the bladder has long been assumed to be sterile; in the meantime, studies have been conducted on the microbiota in the gastrointestinal tract or other body sites of adult or pediatric patients, including those with congenital malformations [1].

Studies using human samples other than urine showed that the microbiota has a key role in the maintenance of health and/or the development of disease. Nowadays, it is widely known that the gut microbiota is of critical importance for homeostasis of the entire body [2], having both local and distal effects. Besides gastrointestinal disorders [3], gut dysbiosis has been linked to neurological or psychiatric disorders, cardiovascular diseases, and metabolic pathologies [4,5]. Similar knowledge has been gathered on the microbiota/microbiome of the oral cavity, skin, respiratory tract, or vagina [6].

Recent advances in molecular biology techniques allowed for us to identify a resident microbial community also in the human urinary tract, previously thought to be sterile in healthy people [7,8,9]. Compared to many other microbial niches in the human body, the urinary tract harbors a relatively low-biomass microbial community, which resides in proximity to other sites with high-biomass microbial communities (i.e., those in the colon or vagina). This could explain the relatively greater difficulties in performing a molecular characterization of the urobiota [8]. Nonetheless, as at other sites, many different bacterial species were found to reside in the urinary tract and form multiple health-associated community structures, called urotypes, with *Lactobacillus* being the most represented microbial genus [9]. It has also been shown how these urotypes differ between men and women or according to age [9].

Like microbial communities at other sites in the human body, local dysbiosis has been associated with disease states. Lower urinary tract functional disorders such as overactive bladder (also known as urgency urinary incontinence (UUI)), interstitial cystitis, chronic prostatitis, chronic pelvic pain syndrome, and bladder pain syndrome were found to be related to changes in the urobiota composition [10]. A possible role of urobiota has also been reported in bladder cancer [11], kidney stone disease [12], and urinary tract infection (UTI) [13].

Spinal dysraphism, also called spina bifida (SB), represents a group of conditions characterized by a neural tube defect originated during embryogenetic development (such as myelomeningocele, myelocele, or meningocele), with a prevalence of about 3 cases per 10,000 pregnancies [14]. Patients with SB may be born with (or develop over time) neurological deficits caused by damage to nerves originating in the lower part of the spinal cord, which include neurogenic bladder, neurogenic bowel, hypoesthesia/anesthesia, and different degrees of lower limb paralysis. In particular, the neurogenic bladder is a condition in which the damaged innervation of the bladder causes the impaired function of the detrusor muscle and the sphincter, resulting in detrusor sphincter dyssynergia. In these patients, neurogenic bladder represents a predisposing factor for UTI and kidney damage [15]. A most frequently used approach to manage this condition is based on clean intermittent catheterization (CIC) and anticholinergic drug use; antibiotic prophylaxis is also frequently used [16]. The associated neurogenic bowel condition also contributes to an increased risk of UTI due to chronic constipation [17,18].

The primary aim of this study is to evaluate any differences in urobiota composition between a cohort of pediatric patients with SB followed up at our center and the general population; the secondary aim is to evaluate any associations between urobiota and the characteristics or treatments of patients with SB.

## 2. Results

### 2.1. Study Population Features

Of the 83 subjects enrolled in the study (44 patients with SB and 39 healthy controls), 34 (40.9%) were females. The mean (±SD) age was 10.0 (4.8) years, whereas the mean (±SD) values of height and weight were 134.0 (24.6) cm and 35.6 (18.9) kg, respectively. Regarding BMI classification, 47 (56.6%) subjects had a normal weight, 14 (16.9%) subjects were underweight, 16 (19.3%) subjects were overweight, and 6 (7.2%) subjects were obese (Table 1).

Table 1 also shows the demographic and clinical characteristics of patients with SB compared to the healthy controls. A statistically significant difference was observed between the two groups in terms of age (*p* < 0.001), weight (*p* = 0.005), and BMI (*p* = 0.001), with patients with SB being older and having higher weight and BMI values than the healthy controls.

Table 2 shows the demographic and clinical characteristics of patients with SB according to whether patients received or did not receive CIC. A statistically significant difference was observed in terms of anticholinergic drug use (100% vs. 50.0%, *p* < 0.001), bowel habits (93.7% vs. 25.0%, *p* < 0.001), and the presence of encopresis (81.2% vs. 16.7%, *p* < 0.001).

Table 3 shows the physical and chemical parameters of urine samples from patients with SB according to whether patients received or did not receive CIC. No statistically significant difference in all listed parameters was observed.

### 2.2. Comparison of Urobiota Features between Patients with SB and Healthy Controls

After excluding four urine samples because of their low-quality reads and insufficient sequencing depth, a total of 4,237,680 reads (from the samples of 43 patients with SB and 36 healthy controls’ samples), corresponding to 1003 unique amplicon sequence variants (ASVs), were available for downstream analyses.

Figure 1A–D depicts the alpha-diversity analysis results for the urine bacterial communities of patients with SB compared to those of the healthy controls. We did not find statistically significant differences in terms of the Shannon diversity index (2.72 ± 1.75 vs. 3.29 ± 1.08; *p* = 0.36) and the inverse Simpson index (17.77 ± 20.94 vs. 19.44 ± 21.44; *p* = 0.29), as well as the Chao1 index (164.02 ± 113.40 vs. 188.97 ± 122.68; *p* = 0.12) and Pielou’s evenness (0.60 ± 0.23 vs. 0.63 ± 0.20; *p* = 0.45). However, statistically significant differences were found in terms of beta diversity, as shown in the Bray–Curtis-distance-based PCoA plots (Figure 1E). Despite a slight overlap between the two groups, samples from patients with SB clustered separately from the samples from healthy controls, as explained by PCoA axis 1 (38.66%) and PCoA axis 2 (22.50%). PERMANOVA confirmed the distinct spatial distribution of samples from the two groups (R^2^ = 0.154; *p* = 0.001). As shown in Appendix A, four main phyla (*Actinobacteria*, *Bacteroidetes*, *Firmicutes*, and *Proteobacteria*) were identified in the urobiota of patients with SB and healthy controls. We found that *Firmicutes* and *Bacteroidetes* were relatively more abundant in the samples of the healthy controls than in the patients with SB (49.8% vs. 36.3% (*p* = 0.006) and 29.7% vs. 14.8% (*p* < 0.001), respectively). Conversely, *Proteobacteria* were relatively more abundant in the samples of patients with SB than in the healthy controls’ samples (35.5% vs. 2.8%, *p* < 0.001).

A genus-level relative abundance analysis was performed to better explain the differences in the urobiota composition between patients with SB and the healthy controls. Network analysis of selected genera allowed for us to observe interactions among bacteria that were apparently stronger in patients with SB than in the healthy controls (Figure 1F,G). We also found (Figure 1H) that, within the phylum *Actinobacteria*, *Bifidobacterium* was relatively more abundant in the samples of patients with SB than in the healthy controls’ samples (9.2% vs. 1.6%, *p* < 0.001). Within the phylum *Bacteroidetes*, abundances of the genera belonging to *Prevotellaceae* were significantly higher in healthy controls’ samples (*p* < 0.001). Interestingly, among uropathogenic bacterial genera, *Enterococcus* and *Escherichia*/*Shigella* were found to be relatively more abundant in the samples of patients with SB than in the healthy controls’ samples (10.8% vs. 1.2% (*p* < 0.001) and 28.6% vs. 3.8% (*p* < 0.001), respectively). A LEfSe-based analysis corroborated these findings, showing that *Enterococcus* and *Escherichia*/*Shigella* were differentially abundant between sample groups (Appendix A).

### 2.3. Comparison of Urobiota Features between SB Patients with or without CIC

Figure 2A–D depicts the alpha-diversity analysis results for the urine bacterial communities of patients with SB with CIC compared to those without CIC. We found statistically significant differences in terms of the Shannon diversity index (2.32 ± 1.83 vs. 3.74 ± 0.99; *p* = 0.03) and the inverse Simpson index (14.46 ± 19.44 vs. 26.32 ± 23.10; *p* = 0.03), as well as the Chao1 index (141.68 ± 120.23 vs. 221.75 ± 67.68; *p* = 0.07) and Pielou’s evenness (0.49 ± 0.29 vs. 0.49 ± 0.17; *p* = 0.04). Statistically significant differences (albeit at a lesser degree) were also found in terms of beta diversity, as shown in the Bray–Curtis-distance-based PCoA plots (Figure 2E). Samples from patients with SB with CIC overlapped with samples from patients with SB without CIC, as explained by PCoA axis 1 (50.63%) and PCoA axis 2 (15.85%). Consistently, PERMANOVA confirmed the scarce spatial distribution of samples from the two groups (R^2^ = 0.063; *p* = 0.04). As shown in Appendix A, we found that the phylum *Bacteroidetes* was significantly less abundant in the samples of patients with CIC than in the samples of patients without CIC (10.7% vs. 18.1%; *p* = 0.02), whereas the phylum *Proteobacteria* was more significantly abundant in the samples of patients with CIC than in the samples of patients without CIC (41.1% vs. 21.2%; *p* = 0.02).

A genus-level relative abundance analysis was performed to better explain the differences in the urobiota composition between patients with SB with or without CIC. Network analysis of the selected genera allowed for us to observe the interactions among bacteria that appeared markedly different between patients with and without CIC (Figure 2F,G). We also found (Figure 2H) that, within the phylum *Actinobacteria*, *Cutibacterium* was relatively more abundant in the samples of patients with CIC than in the samples of patients without CIC (4.9% vs. 1.2%, *p* = 0.01). Within the phylum *Firmicutes*, genera such as *Faecalibacterium*, *Lactobacillus*, *Staphylococcus*, and *Streptococcus* were significantly less abundant in the samples of patients with CIC than in the samples of patients without CIC (*p* = 0.04, *p* = 0.01, *p* = 0.01, and *p* = 0.02, respectively). Regarding the LEfSe-based analysis, these genera were found to be differentially abundant between the sample groups (Appendix A).

## 3. Discussion

Our study is one of the few characterizations of the urobiota (i.e., the bacterial community present in a urine sample) in the pediatric setting published so far and represents, to our knowledge, the first description of the urobiota in pediatric patients with neurogenic bladder. Specifically, this condition was caused by a type of spinal dysraphism (here called SB) such as myelomeningocele in about half of the patients we studied. Of the patients, 73% required CIC and all of them were under treatment with anticholinergic drugs. We showed that urine samples of patients with SB significantly differed from those of the healthy controls regarding not only the overall bacterial community composition but also the relative abundance of specific taxa. These included well-known UTI etiologic agents such as the (*Proteobacteria*) genera *Escherichia*/*Shigella* and the (*Firmicutes*) genus *Enterococcus*, which were more abundant in the samples of patients with SB. Second, patients with CIC had urine samples that significantly differed from those of patients without CIC regarding the predominance (higher in the former) of *Cutibacterium* (*Actinobacteria*), which is a common component of the skin microbiota. Genera such as lactic acid bacteria (i.e., *Lactobacillus*, *Streptococcus*, etc.), which are known to play a protective role against microbial pathogens in the urogenital tract, were less abundant in patients with CIC than in patients without CIC. Taken together, our findings suggest that patients with SB have an altered urobiota related to the neurogenic bladder and that CIC exacerbates the urobiota alterations.

Our findings are reminiscent of data published in 2012 by Fouts et al. [19], who characterized the bacterial communities of urine samples from 53 subjects (aged ≥ 19 years), including 26 healthy controls and 27 patients with spinal cord injury-related neuropathic (i.e., neurogenic in our study) bladder. Of 27 patients (who are equivalent to our 44 patients with SB), 8 voided normally and the remaining 19 (8 and 11, respectively) drained the bladder by intermittent catheterization (i.e., CIC in our study) or indwelling Foley urethral catheterization [19]. Using V1–V3 16S rRNA gene pyrosequencing (which is the technology used at the time that the study was conducted), Fouts et al. [19] showed that neuropathic bladder altered the urobiota, and these alterations consisted into a significantly reduced abundance of specific taxa (i.e., operational taxonomic unit (OTU)-representative reads, which are equivalent to ASVs in our study) such as *Lactobacillus* and *Corynebacterium* (which is another common component of the skin microbiota), and into a significantly increased abundance of other genera such as *Klebsiella*, *Escherichia*, and *Enterococcus*. It is remarkable that, in the Fouts et al.’s study [19], *Lactobacillus* was progressively reduced in abundance (compared to the healthy controls) in neuropathic bladder females who voided the bladder using the clean catch method (as in our study) or who used CIC, and neuropathic bladder females who used indwelling Foley urethral catheterization. This suggests that *Lactobacillus* may be considered a key component of the bacterial community resident in the bladder.

Considering that we analyzed only urine samples presumably derived from the bladder, the decreased relative abundance of *Lactobacilli* and other lactic-acid-producing bacteria in the urobiota of our patients with SB reinforces the role of these microorganisms in the prevention of urinary diseases that affect the pediatric patient, including UTI, UUI/overactive bladder, and urolithiasis [20,21,22,23]. Indeed, while an imbalance or a variation in the urobiota composition may allow for pathogenic bacteria to thrive in the bladder, it has been suggested that the urobiota may have an impact on the outcomes of overactive bladder treatments [24]. In this context, it should be recalled that probiotics, prebiotics, and diet have emerged as alternative treatment strategies for UTI and the other urinary tract diseases mentioned above [25]. None of our patients with SB had urine samples that grew bacterial organisms in culture. This was despite none of our patients had received antibiotics within 30 days before enrolment, or 11 of 32 patients who used CIC had urine samples with esterase detected. Although asymptomatic bacteriuria and pyuria are common in children with neurogenic bladder [26,27,28,29], the management of our patients with culture-negative but esterase-positive urine samples could have been supported from urobiota analysis results to rule-out UTI and, eventually, avoid antibiotic prophylaxis [30]. In the case of patients with symptomatic bacteriuria and pyuria (none of our patients was symptomatic), clinicians could exploit urobiota analysis results to treat symptoms by modifying the urobiota instead of using antibiotics to kill the suspected uropathogen [9].

Here, we tried to overcome the limits of previous urobiota studies [9]. Urine is a low microbial biomass sample, which carries the risk of amplifying contaminant DNA and detecting cross-contamination during the sample collection, preparation, and sequencing steps [31]. To minimize this risk, additional samples (i.e., blank controls) were processed and sequenced along with the study samples. Unlike previous studies that either included small numbers of patients or used different urine sampling methods (i.e., spontaneous midstream voiding, intermittent/permanent catheterization, or suprapubic bladder aspiration) [9], we analyzed a relatively large number of uniquely midstream urine samples obtained from a homogenous cohort of pediatric subjects. We took extreme care when collecting and processing urine samples from our study subjects, particularly with those from patients who have used CIC. Unfortunately, while it is plausible that the urobiota composition in our patients predisposed them to UTI, we were unable to document whether UTI had developed over time in our patients. Finally, this is a single-center study, implying that future studies in other clinical contexts are needed to confirm our findings.

In conclusion, our molecular characterization of the urinary tract bacterial communities of pediatric patients with spinal-dysraphism-induced neurogenic bladder extends the knowledge of the urobiota not only in general but also in association with important urinary tract disorders. The presence in the urine of bacterial species known to cause UTI emphasizes the concept that our efforts to eradicate urinary microbes (typically with antibiotics) lack precision and cause undesired effects on the urobiota. We believe that the information gained with our, or similar, studies provides a solid foundation for increasing the understanding of the pathophysiology of neurogenic bladder and for finding strategies that can improve the management of patients with this limiting condition.

## 4. Material and Methods

### 4.1. Subject Enrolment and Stratification

The urine samples from 83 subjects (44 patients with SB followed up at our center and 39 healthy controls) were collected and analyzed between September 2021 and September 2022. Subjects were enrolled in SB or healthy-control groups according to the following inclusion criteria: (a) aged between 1 and 18 years; (b) no UTI in the previous 30 days; (c) no use of antibiotics in the previous 30 days; (d) no surgery or hospitalization in the last 6 months; (e) no kidney stone disease or congenital anomalies of the kidneys or the urinary tract (CAKUT); (f) state of well-being. All subjects were stratified by age, gender, anthropometric parameters (height, weight, BMI), and bowel habits; patients with SB were also categorized by the type of dysraphism (myelomeningocele or others), degree of mobility (normal or reduced), CIC use, and anticholinergic drug therapy.

### 4.2. Urine Sampling

Urine samples were obtained using the clean catch method, whereby the midstream urine samples from healthy controls and patients with SB were collected into a sterile container. Samples were kept at 4 °C until transported (within an hour) to the clinical microbiology laboratory of the Fondazione Policlinico Universitario A. Gemelli IRCCS (Rome, Italy) for processing. All samples were taken with informed consent and the study was approved by the Regional Committee for Medical Research Ethics (Protocol ID 4279).

### 4.3. Bacterial DNA Isolation from Urine Samples

A 20 mL urine sample was pelleted by centrifugation at 15,000 rpm for 10 min at 4 °C. Supernatant was discarded and the pellet was resuspended in 0.8 mL cetyltrimethylammonium bromide. This suspension was used to extract bacterial DNA with the DANAGENE MICROBIOME Fecal DNA kit (Danagene-Bioted) following the manufacturer’s instruction [32,33,34]. Briefly, 25 µL of proteinase K was added and the suspension was incubated at 70 °C for 10 min. Each sample was transferred in a sterile bead microcentrifuge tube and homogenized for 10 min at room temperature. Finally, 500 µL of the supernatant was transferred to a clear microcentrifuge tube and processed according to the manufacturer’s procedure. Nuclease-free water (Thermo Fisher Scientific, Waltham, MA, USA) containing samples was included and processed as urine samples. DNA from the urine samples was eluted in 70 µL of pre-heated nuclease-free water and stored at −20 °C until processing.

### 4.4. 16S Ribosomal RNA (rRNA) Library Preparation and Sequencing

DNA concentration was assessed using the Qubit 4 fluorometer (Thermo Fisher Scientific) and ds DNA High Sensitivity assay (Thermo Fisher Scientific) according to manufacturer’s procedures. V5–V6 hypervariable regions of the 16S rRNA gene were amplified using the following primers: V5_Next_For: 5′-TCGTCGGCAGCGTCAGATGTGTATAAGAGACAG[ATTAGATACCCYGGTAGTCC]-3′ and V6_Next_Rev: 5′-GTCTCGTGGGCTCGGAGATGTGTATAAGAGACAG[ACGAGCTGACGACA RCCATG]-3′, which contain (from 5′ to 3′) the Nextera transposon sequences and the universal BV5 (Next For) and AV6 (Next Rev) priming sequences [35,36]. A 6 µL aliquot of extracted DNA (diluted at 0.5 µg/mL) was used as the template in a final volume of 50 µL, containing 1 U Phusion High-Fidelity DNA polymerase (Thermo Fisher Scientific), 1X High-Fidelity buffer (Thermo Fisher Scientific), 200 μM dNTPs, and 0.3 μM primers.

The thermal cycling conditions were set as follows: (i) 98 °C for 2 min; (ii) 20 cycles, with 1 cycle consisting of 98 °C for 10 s, 58 °C for 30 s, and 72 °C for 15 s; (iii) 15 cycles, with 1 cycle consisting of 98 °C for 10 s, 62 °C for 30 s, and 72 °C for 15 s; (iv) 72 °C for 7 min. Amplicons were purified using 0.8X Agencourt AMPure XP beads (Beckman Coulter) and eluted into 35 µL of nuclease-free water. Eluted PCR products were quality checked on 1% agarose gel (Thermo Fisher Scientific) and the DNA concentration was measured as previously described [36]. The dual index strategy was used incorporating unique Nextera XT i5 and i7 indexes into both ends of the library molecules. A total of 20 µL of extracted DNA (diluted at 20 ng/mL) was used as the template in a final volume of 50 µL, containing 1U Phusion High-Fidelity DNA polymerase (Thermo Fisher Scientific), 1X High-Fidelity buffer (Thermo Fisher Scientific), 100 μM of dNTPs, 5 μL of i5 index, and 5 μL of i7 index. The thermal cycling conditions were set as follows: (i) 98 °C for 30 s; (ii) 5 cycles, with 1 cycle consisting of 98 °C for 10 s, 63 °C for 30 s, and 72 °C for 3 min. The barcoded amplicons were purified, eluted into 25 µL of nuclease-free water, quality controlled, and the DNA concentration was finally assessed [36]. Each indexed amplicon was equimolarly diluted, and the final pool was prepared for paired-ends sequencing (2 × 250 bp; v2 chemistry; Illumina) on the Illumina MiSeq instrument. To increase the degree of base diversity, the internal control PhiX v3 (Illumina) was added to the DNA library [37].

The sequencing reads were processed using the 16S rRNA gene sequence curation pipeline that was implemented in QIIME2 v2020.6 [38]. Demultiplexing and the quality inspection of reads were performed with QIIME2’s “demux” plugin before the removal of the adapter sequence (5′-CTGTCTCTTATACACATCT-3′) using QIIME2’s “cutadapt trim-paired” plugin. Trimmed sequences were denoised using QIIME2’s “dada2 denoised-paired” plugin [39]. The dechimeric ASVs were summarized with QIIME2’s “feature-table summarize” plugin and classified using QIIME2’s “feature-classifier” plugin [40], which rely on the VSEARCH global consensus alignment against a customized version of the SILVA132 reference database (at 99% sequence similarity). To create a phylogenetic tree of the resulting ASVs, QIIME2’s “phylogeny align-to-tree-mafft-fasttree” plugin was used. Finally, a biological observation matrix (BIOM) was generated merging ASVs’ table and taxonomic information.

### 4.5. Data Analysis

Urobiota data analysis was performed using R v4.0.2 (https://www.rstudio.com/; accessed on 3 March 2023) and the phyloseq package [41]. Unassigned ASVs or the bacterial taxa that were not seen more than two times in at least 5% of samples were removed, thus obtaining reads for a total number of 4,237,680 (median [IQR] number, 46,584). The final dataset was generated applying a taxonomic filtering to remove low prevalent taxa (Epsilonbacteraeota and Deinococcus-Thermus). Samples were normalized to the median sequencing depth (Appendix A) before performing alpha diversity, ordination (beta diversity), and relative abundance analyses. Alpha diversity metrics such as the Shannon diversity index, inverse Simpson diversity index, Chao1 index, and Pielou’s evenness were computed, and the statistical significance was assessed using the Wilcoxon signed rank test.

Beta diversity was evaluated using the Bray–Curtis distance and was graphically represented as principal coordinate analysis (PCoA) plots. We determined the significance between microbial community compositions using the adonis function included in the vegan package, which performs the permutational multivariate analysis of variance (PERMANOVA). Relative abundances were computed at both phylum and genus levels and the statistical significance was assessed by the Wilcoxon signed rank test. Network analysis at the genus level was performed with the phyloseq package functions using the Bray–Curtis distance and by setting the maximum distance at 0.95. To detect changes in the relative abundance at the genus level between the groups of samples, a linear discriminant analysis (LDA) effect size (LEfSe) analysis was performed using the previously recommended parameter settings (i.e., *p* < 0.05; LDA score, >2) [42].

Clinical data were collected using Microsoft Excel 2019 software (version 1808) and analyzed using the IBM SPSS Statistics software. Statistical significance among clinical and demographic variables was assessed using the Welch *t* test and chi-square test, as appropriate. In all analyses, statistical significance was set at a *p* of <0.05.

## Figures and Tables

**Figure 1 ijms-24-08261-f001:**
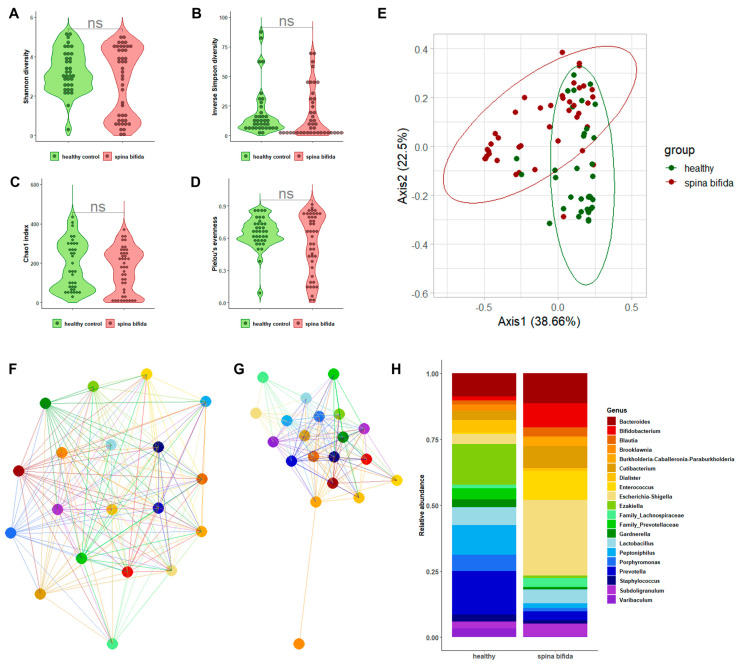
The characterization results of bacterial communities in the urine samples from patients with SB compared to those in the urine samples from healthy controls. Violin plots depict the Shannon diversity index (**A**), inverse Simpson index (**B**), Chao1 index (**C**), or Pielou’s evenness (**D**) alpha-diversity measures in two sample groups. The Bray–Curtis-distance-based PCoA plots depict the spatial separation of two sample groups (**E**). Genus-level network clusters depict bacterial interactions in each of the two sample groups (**F**,**G**). Bar plots depict the relative abundance of 20 bacterial genera in 2 sample groups (**H**).

**Figure 2 ijms-24-08261-f002:**
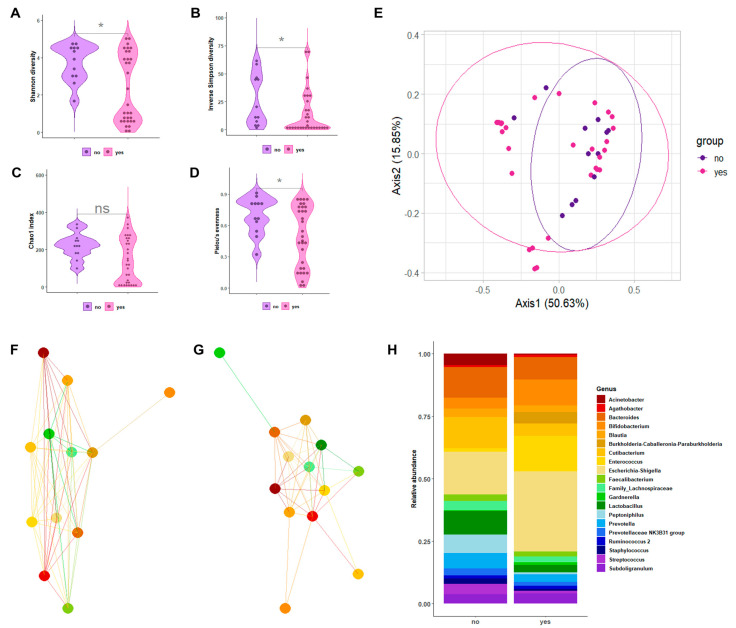
The characterization results of bacterial communities in the urine samples from patients with SB with CIC compared to those in the urine samples from patients with SB without CIC. Violin plots depict the Shannon diversity index (**A**), inverse Simpson index (**B**), Chao1 index (**C**), or Pielou’s evenness (**D**) alpha-diversity measures in two sample groups. Asterisks indicate statistically significant difference between groups. The Bray–Curtis-distance-based PCoA plots depict the spatial separation of two sample groups (**E**). The genus-level network clusters depict bacterial interactions in each of the two sample groups (**F**,**G**). Bar plots depict the relative abundance of 20 bacterial genera in 2 sample groups (**H**).

**Table 1 ijms-24-08261-t001:** Demographic and clinical characteristics of the study population.

	All Subjects (n = 83)	Healthy Controls (n = 39)	SB Patients (n = 44)	*p*
Age (y), mean (SD)	10.0 (4.8)	8.0 (3.5)	11.8 (5.1)	<0.001
Female, n (%)	34 (40.9)	16 (41.0)	18 (40.9)	0.99
Height (cm), mean (SD)	134 (24.6)	129.5 (21.6)	137.9 (26.6)	0.12
Ethnicity, n (%)				0.64
Caucasian	80 (96.4)	38 (97.4)	42 (95.4)	
Asian	1 (1.2)	0 (0.0)	1 (2.3)	
African	2 (2.4)	1 (2.6)	1 (2.3)	
Weight (kg), mean (SD)	35.6 (18.9)	29.7 (14.3)	40.9 (2.0)	0.005
BMI (kg/m^2^), mean (SD)	18.4 (4.6)	16.6 (3.3)	20.0 (5.1)	0.001
BMI classification, n (%)				0.31
Normal weight	47 (56.6)	25 (64.1)	22 (50.0)	
Underweight	14 (16.9)	7 (17.9)	7 (15.9)	
Overweight	16 (19.3)	6 (13.4)	10 (27.7)	
Obesity	6 (7.2)	1 (4.6)	5 (6.4)	

**Table 2 ijms-24-08261-t002:** Demographic and clinical characteristics of patients with SB who received or did not receive CIC.

	No CIC (n = 12)	CIC (n = 32)	*p*
Age (y), mean (SD)	11.5 (6.2)	11.9 (4.7)	0.84
Sex (Female), n (%)	4 (33.3)	14 (43.7)	0.53
Height (cm), n (%)	140.5 (35.0)	137.0 (23.3)	0.75
Ethnicity, n (%)			0.67
Caucasian	12 (100)	30 (93.7)	
Asian	0 (0.0)	1 (3.1)	
African	0 (0.0)	1 (3.1)	
Weight (kg), mean (SD)	46.7 (28.1)	38.7 (17.8)	0.75
BMI (kg/m^2^), mean (SD)	21.0 (5.5)	19.6 (4.9)	0.45
BMI classification, n (%)			0.88
Normal weight	5 (41.7)	17 (53.1)	
Underweight	2 (16.7)	5 (15.6)	
Overweight	3 (25.0)	7 (21.9)	
Obesity	2 (16.7)	3 (9.4)	
Type of spinal dysraphism, n (%)			0.24
Type 1 (myelomeningocele)	4 (33.3)	17 (53.1)	
Type 2 (others)	8 (66.7)	15 (46.9)	
Mobility (independent), n (%)	12 (100)	25 (78.1)	0.08
Anticholinergic drugs, n (%)	6 (50.0)	32 (100)	<0.001
Bowel habits (regular/constipation), n (%)	3 (25.0)	30 (93.7)	<0.001
Constipation treatment, n (%)	1 (8.3)	10 (31.2)	0.16
Encopresis, n (%)	2 (16.7)	26 (81.2)	<0.001

**Table 3 ijms-24-08261-t003:** Urine exam results for patients with SB who received or did not receive CIC.

Parameters Examined	No CIC (n = 12)	CIC (n = 32)	*p*
Turbidity, n (%)	1 (8.3)	0 (0.0)	0.1
Albumin, n (%)	4 (33.3)	8 (25.0)	0.58
Glucose, n (%)	0 (0.0)	0 (0.0)	-
Hemoglobin, n (%)	1 (8.3)	2 (6.2)	0.81
Ketone bodies, n (%)	0 (0.0)	2 (6.2)	0.37
Urobilinogen, n (%)	0 (0.0)	3 (9.4)	0.27
Bilirubin, n (%)	0 (0.0)	0 (0.0)	-
Nitrite, n (%)	0 (0.0)	5 (15.6)	0.15
Esterase, n (%)	0 (0.0)	11 (34.4)	0.06
Specific weight (mean, SD)	1018.2 (7.3)	1018.1 (6.7)	0.98
pH (mean, SD)	5.9 (0.6)	6.0 (0.7)	0.54

## Data Availability

Data are available upon request to the corresponding author.

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
