# Peer review of "Profiling the Urobiota in a Pediatric Population with Neurogenic Bladder Secondary to Spinal Dysraphism"

_ijms, 2023, doi:10.3390/ijms24098261_

Round 1

Reviewer 1 Report

The study is very interesting, I hope to read other studies in the future that delve into this topic.

Author Response

The study is very interesting, I hope to read other studies in the future that delve into this topic.

Answer: I am very grateful to the reviewer for appreciating the study.

Reviewer 2 Report

Congratulations on your work! 

The interest in this field is continuously rising.

There are a few aspects that may be improved in the present paper.

In the discussion section you can include more comparative studies with the adult population.

Strenght and limitations should be specified for the present study.

Fine english errors should be revised.

Author Response

Congratulations on your work! The interest in this field is continuously rising.

Answer: I am very grateful to the reviewer for appreciating the study.

There are a few aspects that may be improved in the present paper.

In the discussion section you can include more comparative studies with the adult population.

Strenght and limitations should be specified for the present study.

Fine english errors should be revised.

Answer: I extensively modified the manuscript, particularly in the Discussion section, which has been enlarged to discuss the results of our study in the context of studies conducted with the adult population. While changing the structure of the Discussion, I added a new paragraph to discuss strengths and limitations of the study. I also revised the English throughout the manuscript. As a result, several sentences have been rewritten.

Reviewer 3 Report

1. The information provided by the analysis of microbial community is too simple, such as α diversity. It is suggested to supplement Simpson index and Chao1 index analysis.

2. Lefse difference analysis is suggested to identify the bacteria taxa makers in the different groups.

3. It would be better if the correlation network of bacteria genus of two groups can be provided respectively.

4. The discussion is not better enough. For example, the study identified significant changes in the abundance of several bacteria. What does this mean? Does it affect patients?

Author Response

  1. The information provided by the analysis of microbial community is too simple, such as α diversity. It is suggested to supplement Simpson index and Chao1 index analysis.

Answer: While thanking the reviewer for her/his suggestion, I have added both Simpson index and Chao 1 index analyses. Accordingly, Figure 1 and Figure 2 have been modified to include new data.

  1. Lefse difference analysis is suggested to identify the bacteria taxa makers in the different groups.

Answer: While thanking the reviewer for her/his suggestion, I have added the results from the LEfSe difference analysis, which are presented in a new Figure (named Supplementary Figure 3).

  1. It would be better if the correlation network of bacteria genus of two groups can be provided respectively.

Answer: While thanking the reviewer for her/his suggestion, I have added the results from the correlation network analysis of bacterial genera for each of the study groups. Accordingly, Figure 1 and Figure 2 have been modified to include new data.

  1. The discussion is not better enough. For example, the study identified significant changes in the abundance of several bacteria. What does this mean? Does it affect patients?

Answer: While thanking the reviewer for her/his suggestion, I have extensively modified the Discussion section to contextualize the results and explain the significance of the observed changes in the abundance of bacterial taxa, as well as to denote strengths and limitations of the study. I have also revised the English throughout the manuscript and several sentences have been rewritten accordingly.

Round 2

Reviewer 3 Report

The manuscript has been improved and is acceptable.